# Multidimensional Markovian BSDEs with Jumps and Continuous Generators

**Mhamed Eddahbi** [1] , **Anwar Almualim** [1,*], **Nabil Khelfallah** [2] **and Imène Madoui** [2]

[1] Department of Mathematics, College of Sciences, King Saud University, P.O. Box 2455, Riyadh 11451, Saudi Arabia
[2] Laboratory of Applied Mathematics, Department of Mathematics, University of Biskra, P.O. Box 145, Biskra 07000, Algeria
* Correspondence: aalmualem@ksu.edu.sa

**Abstract:** We deal with a multidimensional Markovian backward stochastic differential equation driven by a Poisson random measure and independent Brownian motion (BSDEJ for short). As a first result, we prove, under the Lipschitz condition, that the BSDEJ's adapted solution can be represented in terms of a given Markov process and some deterministic functions. Then, by means of this representation, we show existence results for such equations assuming that their generators are totally or partially continuous with respect to their variables and satisfy the usual linear growth conditions. The ideas of the proofs are to approximate the generator by a suitable sequence of Lipschitz functions via convolutions with mollifiers and make use of the $L^2$–domination condition, on the law of the underlying Markov process, for which several examples are given.

**Keywords:** Markov processes; backward stochastic differential equation with jumps; Poisson random measure; Brownian motion

**MSC:** 60G44; 60G57; 60H10; 60J25



## 1. Introduction

Let $\{\Omega, \mathfrak{F}, \mathbf{P}\}$ be a complete probability space, $\{\mathfrak{F}_t\}_{t \geq 0}$ be a nondecreasing family of sub-$\sigma$-algebras of $\mathfrak{F}$, on which are defined two fundamental time homogeneous independent stochastic processes: a standard $\mathbb{R}^q$-valued Wiener $\{W_t : t \in [0, T]\}$ and a real-valued Poisson random measure $N(\mathrm{d}s, \mathrm{d}e)$ defined in $[0, T] \times E$, where $E = \mathbb{R}^q \setminus \{0_{\mathbb{R}^q}\}$. We denote also by $\mathrm{d}s\nu(\mathrm{d}e)$ the *compensator* of $N$, in other words:

$$\widetilde{N}(\mathrm{d}s, \mathrm{d}e) = N(\mathrm{d}s, \mathrm{d}e) - \nu(\mathrm{d}e)\mathrm{d}s,$$

$\widetilde{N}$ is a martingale with mean zero called the *compensated* Poisson random measure. For the theory of stochastic differential equations with Poisson's measure we refer to [1].

We consider $\mathbb{F} = (\mathfrak{F}_t)_{t \in [0,T]}$ to be the filtration generated by the two processes $W$ and $\widetilde{N}$. In this work, we are interested in the following backward stochastic differential equation driven by both a Wiener and a Poisson random measure. For a given $\mathbb{R}^q$-valued random variable $\zeta$ defined on $(\Omega, \mathfrak{F}_T, \mathbf{P})$ and an $\mathbb{R}^p$-valued càdlàg Markov process $(X_t)_{t \in [0,T]}$ on $(\Omega, \mathfrak{F}_T, \mathbb{F}, \mathbf{P})$, we consider the following multidimensional BSDEJ: for any $t \in [0, T]$

$$\begin{aligned} Y_t &= \xi + \int_t^T f(r, X_r, Y_r, Z_r, K_r(\cdot))\mathrm{d}r \\ &\quad - \int_t^T Z_r \, \mathrm{d}W_r - \int_t^T \int_E K_r(e)\widetilde{N}(\mathrm{d}r, \mathrm{d}e). \end{aligned} \tag{1}$$

Recall that BSDE (1) without the jump part was first studied by Pardoux and Peng [2], whereby they addressed the existence and uniqueness problem under the globally Lipschitz

condition. Since then, the theory of BSDE has known enormous growth and has been applied to several domains such as stochastic control and optimization, game theory, mathematical finance, economics, partial differential equations, etc. For more information, we refer the reader to [3,4] and the references therein. Specifically, among these extensions we mention that Tang and Li [5] were the first authors who studied BSDES driven by a Poisson random measure and independent Brownian motion of the type BSDEJ (1). They proved the existence of a unique solution for such equations under the Lipschitz conditions.

There is a huge literature devoted to the theory of one-dimensional BSDEs driven by Brownian motion with continuous generators. Firstly, Lepeltier and Martin [6] proved the existence of a solution for such BSDEs when the driver satisfies the linear growth condition, and the terminal condition is square integrable. Later, Jia and Peng [7], based on the result found in [6], showed that underlying BSDE has either one or uncountably many solutions. They also provided the structure of those solutions. Then, Kobylanski [8] provided existence, comparison, and stability results for one-dimensional continuous BSDEs with a quadratic growth in the Brownian component and the terminal condition is assumed to be bounded. Finally, Fan and Jiang [9] discussed the existence of the minimal solution to BSDE whose generator satisfies linear growth conditions in $(y, z)$, left-continuous and lower semi-continuous in $y$ and continuous in $z$. Compared to the continuous setting there were few papers dealing with the existence problem for BSDE with jumps and less regular coefficients. Yin and Mao [4] dealt with a class of one-dimensional BSDE with Poisson jumps and with random terminal times. They showed the existence and uniqueness of a minimal solution for BSDE whose driver has a linear growth. Then, Qin and Xia [10] proved the existence of a minimal solution for BSDEs driven by Poisson processes where the coefficient is continuous and satisfies an improved linear growth assumption. They also extended the result to the case where the coefficient is left or right continuous. More recently, Madoui et al. [11] and Abdelhadi et al. [12] provided some examples that ensure the connection between one type of quadratic BSDEs with jumps and standard BSDEs with continuous drivers. It is worth pointing out that all the previously mentioned results are given for one-dimensional BSDE and the main tools in the proofs are approximating technique and the comparison theorem.

To the best of our knowledge, the first result dealing with multidimensional BSDE with continuous generator was provided by Hamadène [13]. The author obtained the existence of a solution for multidimensional BSDE under the assumptions that the generator $f$ is uniformly continuous with respect to $y$, $z$ and the $i$th component $f_i$ of $f$ depends only on the $i$th row of $z$. Secondly, Hamadène and Mu [14], via an existence result for a multidimensional Markovian BSDE with continuous coefficient and stochastic linear growth, proved the existence of Nash equilibrium point for a non-zero-sum stochastic differential games. Subsequently, the result was extended to a coupled system of BSDEs in Mu and Wu [15]. Our results come to complete these studies in the setting of BSDEs with jumps and continuous generators.

Since the aim of the first result of this paper is to investigate a deterministic representation theorem for the solution of the Equation (1). We first recall some existing results in the literature that study regularity and representation of the viscosity solution of partial differential equations via the solution of forward–backward stochastic differential equation driven by continuous Brownian motion. The first result that went in this direction was established by Pardoux and Peng in [2] which claims the backward components $Y$ can be determined in terms of the forward component $X$, when the coefficients satisfy the Lipschitz continuity condition. Then, they proved, under more strong smoothness conditions on the coefficients (e.g., of class $\mathcal{C}^3$ in their spacial variables), that the Brownian component $Z$ has continuous paths. Two years later, under the previous smoothness conditions, Ma et al. [16] proved the following explicit representation for all $s$ in $[t, T]$, $Y_s^{t,x} = u(s, X_s^{t,x})$ and $Z_s^{t,x} = \partial_x u(s, X_s^{t,x}) \sigma(s, X_s^{t,x})$. Subsequently, this result was weakened by Ma and Zhang [17] where they relaxed the smoothness condition on the coefficients by assuming that they are only $\mathcal{C}^1$ and the diffusion coefficient of the forward component

is uniformly elliptic. Later, N'zi et al. [18] studied the regularity of the viscosity solution of a quasi-linear parabolic partial differential equation with merely Lipschitz coefficients. The main results are obtained by using Krylov's inequality in the case, where the diffusion coefficient of the forward equation is uniformly elliptic. In the degenerate case, they exploited the idea used by Bouleau-Hirsch on absolute continuity of probability density measures. On the other hand, when the Markov process is a solution of some SDE with jumps, it is shown in Barles et al. [19] that the solution $Y_s^{t,x}$ of a class of BSDE with jumps provides a viscosity solution of PIDE by mean the deterministic functions $u(t, x) = Y_t^{t,x}$ but no representation has been given for the $Z_s^{t,x}$ and $K_s^{t,x}(\cdot)$.

As the first result of this paper, under standard assumptions where the generator $f$ of BSDEJ (1) satisfies the Lipschitz and of the linear growth conditions, we prove, without using the connection with PIDE, a new representation theorem for BSDEs with jumps (Theorem 1). This is performed with less regularity on the generator and the Markov process. Based on the seminal paper on semi-martingale theory by Çinlar et al. [20] and Çinlar and Jacod [21], we represent the components of the adapted solution in terms of the Markov process $(X_s^{t,x})_{s \in [0,T]}$ starting at $x$ at time $t$. In other words, we prove the existence of three deterministic functions $u$, $v$ and $\theta$ such that for all $s$ in $[t, T]$, $Y_s^{t,x} = u(s, X_s^{t,x})$, $Z_s^{t,x} = v(s, X_s^{t,x})$ and $K_s^{t,x}(\cdot) = \theta(s, X_{s-}^{t,x}, \cdot)$. In fact, this result generalizes the one obtained by El Karoui et al. [3] to the jump case.

As the second result, starting with the case where $f$ satisfies the linear growth condition, with the help of the $L^2$-domination property and some lower and upper bounds of the density of the law related to the transition probability of the underlying Markov process, we prove an existence result for BSDEJ (1) with a continuous generator in $y$, $z$, and globally Lipschitz in $k(\cdot)$ such that $\int_0^T \mathbb{E}|Y_r|^2 \mathrm{d}r < \infty$. Then, for the case where $f$ satisfies the sub-linear growth condition, we prove that $\mathbb{E} \sup_{0 \le t \le T} |Y_t|^2 < \infty$. Finally, by assuming that the generator $f$ depends on $x$, $y$, $z$, and $\int_0^T k(e)\nu(\mathrm{d}e)$ rather than $k(\cdot)$, we obtain the existence of at least one solution to BSDEJ whose generator is continuous in $y$, $z$, and $k$. Notice that our results use neither a comparison theorem, nor a deterministic representation usually obtained by partial integral differential equations.

This paper is divided into four sections. In Section 2, we shall give some preliminaries, introduce some notations and definitions and state some technical results. Section 3 deals with the deterministic representation of solutions of BSDEJ by means of the representation of additive functionals of Markov processes to establish the existence and uniqueness of solutions of our BSDEJ in the Markovian case.

In Section 4, we deal with Markovian BSDEJ with a continuous generator and prove the existence of at least one solution to our Markovian BSDEJ using the so-called $L^2$-domination technique and some regularization and approximation arguments. Furthermore, some special cases on linear and sub-linear growth conditions and the regularity of the generator are discussed. We conclude this paper with several examples of Markov processes with the $L^2$-domination property.

## 2. Preliminaries and Auxiliary Results

In this section, we collect some technical results that will be needed in the proofs of our main results along different sections of the paper. We start by providing some definitions and notations.

- For any $x \in \mathbb{R}^q$, $|x|^2 = \sum_{i=1}^q x_i^2$ denotes its Euclidean norm.
- $L^2(\Omega, \mathfrak{F}, \mathbf{P}, \mathbb{R}^q)$: the Banach space of $\mathbb{R}^q$-valued, square-integrable random variables on $(\Omega, \mathfrak{F}, \mathbb{P})$.
- $\mathcal{M}_{\mathfrak{F}}^2(0, T, \mathbb{R}^q)$: the Banach space of $\mathbb{R}^q$-valued $\mathfrak{F}_t$-adapted processes $\varphi_\cdot$ such that

$$\int_0^T \mathbb{E}|\varphi_t|^2 \mathrm{d}t < \infty.$$

- $\mathcal{L}_\nu^{2,q} := \mathcal{L}^2(E, \mathbb{R}^q, \nu(de))$: the Banach space of $\mathbb{R}^q$-valued deterministic functions $(\varphi(e))_{e \in E}$ such that

$$\|\varphi(\cdot)\|_{q,\nu}^2 = \int_E |\varphi(e)|^2 \nu(de) < \infty.$$

- $\mathcal{M}_{\mathfrak{F}}^2([0, T] \times E, \mathbb{R}^q, dt\nu(de))$: the Banach space of $\mathbb{R}^q$-valued $\mathcal{F}_t$-adapted processes $(\psi_t(e))_{0 \leq t \leq T, e \in E}$ such that

$$\int_0^T \mathbb{E}\|\psi_t(\cdot)\|_{q,\nu}^2 dt = \int_0^T \int_E \mathbb{E}|\psi_t(e)|^2 \nu(de) dt < \infty.$$

- $\mathcal{S}_{\mathfrak{F}}^2(0, T; \mathbb{R}^q)$: the Banach space of $\mathbb{R}^q$-valued, $\mathfrak{F}_t$-adapted, and càdlàg processes $(Y_t)_{0 \leq t \leq T}$ such that

$$\mathbb{E} \sup_{0 \leq t \leq T} |Y_t|^2 < \infty.$$

- For the convenience of notations we set:

$$\mathbb{M}^2 = \mathcal{M}_{\mathfrak{F}}^2(0, T, \mathbb{R}^q) \times \mathcal{M}_{\mathfrak{F}}^2(0, T, \mathbb{R}^{q \times q}) \times \mathcal{M}_{\mathfrak{F}}^2([0, T] \times E, \mathbb{R}^q, dt\nu(de)),$$

and

$$\mathbb{M}_S^2 = \mathcal{S}_{\mathfrak{F}}^2(0, T; \mathbb{R}^q) \times \mathcal{M}_{\mathfrak{F}}^2(0, T, \mathbb{R}^{q \times q}) \times \mathcal{M}_{\mathfrak{F}}^2([0, T] \times E, \mathbb{R}^q, dt\nu(de)).$$

*Representation of Additive Functionals of Markov Processes*

Let $X = (\Omega, \mathfrak{F}, \mathfrak{F}_t, \theta_t, X_t, \mathbf{P}_x)$ be a right-continuous left-hand limited strong Markov process with an infinite lifetime, with state space $\mathbb{R}^p$. The operators $\theta_t$, $t \geq 0$, are called the *shift operators* defined by

$$X_s(\theta_t(\omega)) = X_{t+s}(\omega),$$

where as usual $X$ is the coordinate process.

Assume further, along the rest of the paper, that $X$ is *a right process* in the sense of Getoor see ([22], [(9.7) Terminology p. 55]).

**Definition 1.** *(i) An additive locally square integrable martingale on $(X, (\mathfrak{F}_t)_{0 \leq t \leq T})$ is an $\mathbb{R}^p$-valued process $Y$ that is adapted to $(\mathfrak{F}_t)_{0 \leq t \leq T}$, is right-continuous, is a locally square integrable local martingale on $(\Omega, \mathfrak{F}, (\mathfrak{F}_t)_{0 \leq t \leq T}, \mathbf{P}_x)$ for every $x \in \mathbb{R}^p$ and is additive with respect to $(\theta_t)$ (vanishing at 0), and for every pair $(t, u)$,*

$$Y_{t+u} = Y_t + Y_u \circ \theta_t$$

*almost surely.*
*(ii) We say that $Y$ is quasi-left-continuous if $Y_{T_n} \longrightarrow Y_T$ almost surely for every increasing predictable sequence $(T_n)_{n \geq 0}$ of stopping times with finite limit $T$.*

First, we recall some more facts about semi-martingales which are defined on the probability space $(\Omega, \mathfrak{F}, (\mathfrak{F}_t)_{t \geq 0}, \mathbf{P}_x)$. We consider a $q$-dimensional semi-martingale $Y_\cdot = (Y_\cdot^i)_{1 \leq i \leq q}$. We define the $q$-dimensional process $Y_\cdot^e = ((Y_\cdot^e)^i)_{1 \leq i \leq q}$

$$Y_t^e = \sum_{0 < s \leq t} \Delta Y_s \mathbb{1}_{\{|\Delta Y_s| \geq 1\}},$$

where $\mathbb{1}_G$ stands for the indicator function of the set $G$ and $\Delta Y_s = Y_s - Y_{s-}$. It is well known that $Y_\cdot^e$ is a right-continuous pure jump process which has finitely many jumps in any finite interval. Therefore, the semi-martingale $Y_\cdot - Y_\cdot^e$ has bounded jumps and can be decomposed uniquely as follows

$$Y_t - Y_t^e = Y_0 + Y_t^b + Y_t^c + Y_t^d,$$

where $Y^b$ is a predictable process of bounded variation on every finite interval $Y^c$ is a continuous local martingale, and $Y^d$ is a purely discontinuous local martingale (corresponding to the a compensated sum of jumps). Moreover, $Y_0^c = Y_0^d = 0$. The canonical decomposition of the $q$-dimensional special semi-martingale $Y.$ is

$$Y_t = Y_0 + Y_t^b + Y_t^c + Y_t^d + Y_t^e. \tag{2}$$

In fact, the decomposition (2) is unique up to a $\mathbf{P}_x$-null set. All the above processes are $q$-dimensional, for example, the $i$th component of $Y_t^c$ is simply $Y_t^{ic}$.

We define the following integer-valued random measure $\Gamma$ on $\mathbb{R}_+ \times \mathbb{R}^q$ by

$$\Gamma(\omega, \mathrm{d}t, \mathrm{d}y) = \sum_{s>0} \mathbf{1}_{\{\Delta Y_s(\omega) \neq 0\}} \delta_{(s, \Delta Y_s(\omega))}(\mathrm{d}t, \mathrm{d}y),$$

$\Gamma$ is called the jump measure of $Y$.

Let $B_t = Y_t^b$ in the decomposition (2) $C_t = (C_t^{ij})_{1 \leq i,j \leq q} = (\langle Y_\cdot^{ic}, Y_\cdot^{jc} \rangle_t)_{1 \leq i,j \leq q}$ and $\gamma$ is the dual predictable projection of $\Gamma$ (called also the compensator).

The triplet $(B, C, \gamma)$ is called the *local characteristics* of $Y$ which is unique, up to a $\mathbf{P}$-null set. In fact, one can choose a version of $(B, C, \gamma)$ that satisfies the following conditions:

**a.** For all $t \geq s \geq 0$, $C_t - C_s$ is a non-negative symmetric matrix;

**b.** $\Gamma(\omega, \mathbb{R}_+ \times \{0_{\mathbb{R}^q}\}) = 0$;

**c.** $\int_{\mathbb{R}^q} (1 \wedge |y|^2) \gamma(\omega, [0,t], \mathrm{d}y) < \infty$ for every $t \geq 0$.

According to ([21], [Theorem 2.43]), a $q$-dimensional additive semi-martingale has the decomposition (2), Moreover, $B.$, and $C.$ are $\mathbb{F}$-predictable additive processes and $\gamma$ is an $\mathbb{F}$-predictable additive random measure.

**Lemma 1** ([21], Theorem 2.44). *Let $Y$ be a $q$-dimensional additive semi-martingale on $\left(\Omega, (\mathfrak{F}_t)_{t \geq 0}, \mathbf{P}\right)$ which is quasi-left-continuous. Then, there exist:*

**(i)** *An $(\mathfrak{F}_t)$-adapted continuous increasing additive functional $A$;*

**(ii)** *An $\mathcal{B}(\mathbb{R}^q)$-measurable $\mathbb{R}^q$-valued function $b = (b_1, \ldots, b_q)$;*

**(iii)** *An $\mathcal{B}(\mathbb{R}^{q \times q})$-measurable $\mathbb{R}^{q \times q}$-valued function lower triangular matrix-valued function $c = (c_{ij})_{1 \leq i,j \leq q}$ of measurable functions such that $c_{ij} = 0$ if $j > i$, or if $c_{jj} = 0$;*

**(iv)** *A positive kernel $\Theta(x, \mathrm{d}y)$ from $(\mathbb{R}^q, \mathcal{B}(\mathbb{R}^q))$ to $(\mathbb{R}^q, \mathcal{B}(\mathbb{R}^q)$ having*

*$\Theta(x, \{0_{\mathbb{R}^q}\}) = 0$ for all $x \in \mathbb{R}^q$ such that*

$$\int_{\mathbb{R}^q} f(x, y) \Theta(x, \mathrm{d}y) < \infty \text{ for all } x \in E. \tag{3}$$

*for $\mathcal{B}(\mathbb{R}^q) \otimes \mathcal{B}(E)$-measurable strictly positive function $f$.*

*Such that*

$$B_t = \int_0^t b(X_s) \mathrm{d}A_s, \quad C_t = \int_0^t cc^*(X_s) \mathrm{d}A_s \quad and \quad \gamma(\mathrm{d}s, \mathrm{d}y) = \Theta(X_s, \mathrm{d}y) \mathrm{d}A_s,$$

*define a version $(B, C, \gamma)$ of the triplet of local characteristics of $Y$ under every $\mathbf{P}_x$, $x \in \mathbb{R}^p$.*

Consider the following assumptions:

**($\mathbf{A_1}$)** Let $Y = (Y^i)_{1 \leq i \leq q}$ be a collection of continuous additive local martingales, on $\left(\Omega, (\mathfrak{F}_t)_{t \geq 0}\right)$ such that $\mathrm{d}\langle Y_\cdot^i, Y_\cdot^i \rangle_t \ll \mathrm{d}t$ almost surely, for all $1 \leq i \leq q$.
Let $c = (c_{ij})_{1 \leq i,j \leq q}$ be the collection of $\mathcal{B}(\mathbb{R}^{q \times q})$-measurable functions whose existence and properties are given by the Lemma 1 with $A_t = t$;

($A_2$) Let $\Gamma$ be an additive integer-valued random measure on $\mathbb{R}_+ \times \mathbb{R}^q$ defined over $(\Omega, (\mathfrak{F}_t)_{t \geq 0})$. Let $\gamma$ be its dual predictable projection. For each $G$ in $\mathcal{B}(\mathbb{R}^q)$ and $t > 0$ set $\gamma_t^G = \gamma([0, t] \times G)$. Assume that $\mathrm{d}\gamma_t^G \ll \mathrm{d}t$ almost surely such that the mapping $t \longmapsto \gamma_t^G$ is locally integrable. This is equivalent to the existence of a positive kernel $\Theta(x, \mathrm{d}y)$ on $(\mathbb{R}^q, \mathcal{B}(\mathbb{R}^q))$ satisfying (3) and $\gamma(\mathrm{d}s, \mathrm{d}y) = \Theta(X_s, \mathrm{d}y)\mathrm{d}s$ a.s.

**Lemma 2** ([21], Lemma 3.4 and Theorem 3.7). *Under the assumptions* ($A_1$) *and* ($A_2$), *there exists a Wiener process and a Poisson random measure both still denoted by* $W = (W^i)_{1 \leq i \leq q}$ *and $N$ on $\mathbb{R}_+ \times \mathbb{R}^q$ with compensator $\mathrm{d}s\nu(\mathrm{d}e)$ (by extending the probability space if necessary by usual product spaces) such that*

$$Y_t^i = \sum_{j=1}^q \int_0^t c_{ij}(X_s)\mathrm{d}W_s^j \ \text{for all} \ i = 1, 2, \ldots, q,$$

*and*

$$\Gamma(G) = \int\int_{\mathbb{R}_+ \times E} \mathbb{1}_G(s, \theta(X_{s-}, e))N(\mathrm{d}s, \mathrm{d}e) \ \text{for all} \ G \in \mathcal{B}(\mathbb{R}_+) \otimes \mathcal{B}(\mathbb{R}^q),$$

*where $\theta$ is a measurable function satisfying*

$$\Theta(x, H) = \int_E \mathbb{1}_H(\theta(x, e)\nu(\mathrm{d}e)) \ \text{for all} \ x \in \mathbb{R}^p \ \text{and for all} \ H \in \mathcal{B}(\mathbb{R}^q).$$

**Remark 1.** *Suppose $(X_t)_{0 \leq t \leq T}$ is a semi-martingale Markov process on $\mathbb{R}^p$ that is not time-homogeneous, then the time-homogeneous process $(t, X_t - X_0)$ is an $\mathbb{R}^{p+1}$-valued semi-martingale additive functional. Therefore, the measurable functions $b_i(x)$, $c_{ij}(x)$, and $\theta(x, e)$ become $b_i(s, x)$, $c_{ij}(s, x)$, and $\theta(s, x, e)$.*

### 3. Deterministic Representation for Markovian BSDEJ

In this section, we are interested in a class of multidimensional BSDE with jumps for which the generator $f$ and the random terminal value $\xi$ at time $T$ are both functions of a *right process $X$* on the filtered probability space $(\Omega, \mathfrak{F}_T, (\mathfrak{F}_t)_{t \in [0,T]}, \mathbf{P}_x)$ for $x \in \mathbb{R}^p$. Notice that the filtration $(\mathfrak{F}_t)_{t \in [0,T]}$ is generated by the Markov process $X$ and two processes obtained in the Lemma 2, still denoted $W$ and $\widetilde{N}$.

Our objective is to generalize to the jump case the work of El Karoui et al. ([3], Theorem 4.1, p. 46). That is, to represent the components of the BSDEJ's adapted solution in Lipschtiz framework in terms of $X$ and some deterministic functions. Compared with the representation by the well-known Feynman–Kac formula using PIDEs, our method does not require regularity on the coefficients.

From now, we shall deal with the following Markovian BSDEJ: for all $t \leq s \leq T$ and $x \in \mathbb{R}^p$

$$
\begin{aligned}
Y_s^{t,x} \ = \ & g(X_T^{t,x}) + \int_s^T f(r, X_r^{t,x}, Y_r^{t,x}, Z_r^{t,x}, K_r^{t,x}(\cdot))\mathrm{d}r \\
& - \int_s^T Z_r^{t,x}\,\mathrm{d}W_r - \int_s^T \int_E K_r^{t,x}(e)\widetilde{N}(\mathrm{d}r, \mathrm{d}e),
\end{aligned}
\tag{4}
$$

where $(X_s^{t,x})_{s \geq 0}$ is an $\mathbb{R}^p$-valued *right process* and $X_s^{t,x} = x$ if $s \leq t$.

The following assumptions will be considered in this paper.

$$f : [0, T] \times \mathbb{R}^p \times \mathbb{R}^q \times \mathbb{R}^{q \times q} \times \mathcal{L}_\nu^{2,q} \longrightarrow \mathbb{R}^q, \ g : \mathbb{R}^p \longrightarrow \mathbb{R}^q$$

are measurable functions, and satisfy the following hypotheses:

($H_{3.1}$) $\sup_{0 \leq s \leq T} \mathbb{E}[|X_s^{t,x}|^2] < \infty$.

(**H**$_{3.2}$)　For any $(r, x, y, z) \in [0, T] \times \mathbb{R}^p \times \mathbb{R}^q \times \mathbb{R}^{q \times q}$ and $k \in \mathcal{L}_\nu^{2,q}$, $|g(x)| \leq C(1 + |x|)$ and

$$|f(r, x, y, z, k(\cdot))| \leq C(1 + |x| + |y| + |z| + \|k(\cdot)\|_{q,\nu}).$$

(**H**$_{3.3}$)　There exists $L \geq 0$, such that for all $r \in [0, T]$, $\forall\, x \in \mathbb{R}^p$, $\forall\, (y, y') \in \mathbb{R}^q$, $\forall\, (z, z') \in \mathbb{R}^{q \times q}$ and $(k(\cdot), k'(\cdot)) \in \mathcal{L}_\nu^{2,q}$

$$\left| f(r, x, y, z, k(\cdot)) - f(t, x, y', z', k'(\cdot)) \right| \leq L(|y - y'| + |z - z'| + \left\| (k - k')(\cdot) \right\|_{q,\nu}).$$

In view of the hypotheses (**H**$_{3.1}$), (**H**$_{3.2}$), and (**H**$_{3.3}$) the Lemma 2.4 in [5], see also [23–26] among others, confirms that the BSDEJ (4) admits a unique solution $(Y_s^{t,x}, Z_s^{t,x}, K_s^{t,x}(\cdot))_{s \leq T}$ which belongs to $\mathbb{M}^2$.

In what follows, we are interested to establish a Markovian structure of the solution $(Y_s^{t,x}, Z_s^{t,x}, K_s^{t,x}(\cdot))_{s \leq T}$ of a BSDEJ in terms of some deterministic measurable functions evaluated at $(s, X_s^{t,x})$.

The following Lemma is found to be useful.

**Lemma 3.** *Under the assumptions* (**H**$_{3.1}$)*–*(**H**$_{3.3}$)*. There exists a constant C such that for any $t \leq s \leq T$, we have*

$$\mathbb{E}\left[ |Y_s^{t,x}|^2 + \int_0^T \left( |Z_r^{t,x}|^2 + \|K_r^{t,x}(\cdot)\|_{q,\nu}^2 \right) dr \right] \leq C\left( 1 + |x|^2 \right). \tag{5}$$

**Proof.** Applying Itô's formula from $s$ to $T$, to $|y|^2$ with the Equation (4), we obtain

$$
\begin{aligned}
& |Y_s^{t,x}|^2 + \int_s^T \left( |Z_r^{t,x}|^2 + \|K_r^{t,x}(\cdot)\|_{q,\nu}^2 \right) dr \\
= {}& \left| g(X_T^{t,x}) \right|^2 + 2 \int_s^T Y_r^{t,x}\, f(r, X_r^{t,x}, Y_r^{t,x}, Z_r^{t,x}, K_r^{t,x}(\cdot)) dr \\
& - (M_T^{t,x} - M_s^{t,x}) - (N_T^{t,x} - N_s^{t,x}),
\end{aligned}
$$

where

$$M_s^{t,x} = 2 \int_0^s Y_r^{t,x} Z_r^{t,x}\, dW_r + 2 \int_0^s \int_E Y_r^{t,x} K_r^{t,x}(e) \widetilde{N}(dr, de)),$$

and

$$N_s^{t,x} = \int_0^s \int_E |K_r^{t,x}(e)|^2 \widetilde{N}(dr, de)),$$

are real-valued martingales.

If we take the expectation in each member, we obtain

$$
\begin{aligned}
& \mathbb{E}\left[ |Y_s^{t,x}|^2 + \int_s^T \left( |Z_r^{t,x}|^2 + \|K_r^{t,x}(\cdot)\|_{q,\nu}^2 \right) dr \right] \\
= {}& \mathbb{E}\left| g(X_T^{t,x}) \right|^2 + 2\mathbb{E}\left[ \int_s^T Y_r^{t,x}\, f(r, X_r^{t,x}, Y_r^{t,x}, Z_r^{t,x}, K_r^{t,x}(\cdot)) dr \right].
\end{aligned}
$$

Making use of the linear growth of $g$ and $f$ and $|ab| \leq \varepsilon |a|^2 + \dfrac{1}{\varepsilon}|b|^2$ for any $\varepsilon > 0$ and $a, b \in \mathbb{R}^q$, we obtain, by the usual techniques for BSDEs,

$$
\begin{aligned}
& \mathbb{E}\left[ |Y_s^{t,x}|^2 + \frac{1}{2} \int_s^T \left( |Z_r^{t,x}|^2 + \|K_r^{t,x}(\cdot)\|_{q,\nu}^2 \right) dr \right] \\
\leq {}& C\left( 1 + \mathbb{E}\left| X_T^{t,x} \right|^2 + \int_s^T \mathbb{E}|X_r^{t,x}|^2 dr + \int_s^T \mathbb{E}|Y_r^{t,x}|^2 dr \right).
\end{aligned}
\tag{6}
$$

It follows, thanks to Gronwall's Lemma and Assumption ($\mathbf{H}_{3.1}$), that

$$\mathbb{E}|Y_s^{t,x}|^2 \le C\left(1 + \sup_{0 \le s \le T} \mathbb{E}|X_s^{t,x}|^2\right) \le C(1 + |x|^2).$$

Similarly, from (6), one can arrive at

$$\mathbb{E}\left[\int_s^T \left(|Z_r^{t,x}|^2 + \|K_r^{t,x}(\cdot)\|_{q,\nu}^2\right)\mathrm{d}r\right] \le C(1 + |x|^2).$$

Finally, a combination of the two above inequalities leads to (5), which achieves the proof.  □

**Theorem 1.** *Under the assumptions* ($\mathbf{H}_{3.1}$)–($\mathbf{H}_{3.3}$)*, there exist three measurable and deterministic functions* $u : [0, T] \times \mathbb{R}^p \longrightarrow \mathbb{R}^q$*,* $v : [0, T] \times \mathbb{R}^p \longrightarrow \mathbb{R}^{q \times q}$ *and* $\theta : [0, T] \times \mathbb{R}^p \longrightarrow \mathcal{L}_\nu^{2,q}$ *such that for any* $(s, e) \in [t, T] \times E$

$$Y_s^{t,x} = u(s, X_s^{t,x}), \quad Z_s^{t,x} = v(s, X_s^{t,x}) \quad \text{and} \quad K_s^{t,x}(e) = \theta(s, X_{s-}^{t,x}, e).$$

*Moreover,* $\forall\, (s, x) \in [t, T] \times \mathbb{R}^p$,

$$u(s, x) = \mathbb{E}\left[g(X_T^{s,x}) + \int_s^T f(r, X_r^{s,x}, Y_r^{s,x}, Z_r^{s,x}, K_r^{s,x}(\cdot))\mathrm{d}r\right].$$

*and is continuous such that* $|u(t, x)| \le C(1 + |x|)\ \forall\, (t, x) \in [0, T] \times \mathbb{R}^p$.

**Proof.** We split it up into two steps.
**Step 1.** In this step, we suppose that $f$ does not depend on $y$, $z$ and $k(\cdot)$, in which case, Equation (4) becomes

$$Y_s^{t,x} = g(X_T^{t,x}) + \int_s^T f(r, X_r^{t,x})\mathrm{d}r - \int_s^T Z_r^{t,x}\, \mathrm{d}W_r - \int_s^T \int_E K_r^{t,x}(e)\widetilde{N}(\mathrm{d}r, \mathrm{d}e) \tag{7}$$

Therefore, by taking the conditional expectation with respect to $\mathfrak{F}_s$, we obtain for all $t \le s \le T$

$$\begin{aligned}
Y_s^{t,x} &= \mathbb{E}\left[g(X_T^{t,x}) + \int_s^T f(r, X_r^{t,x})\mathrm{d}r \mid \mathfrak{F}_s\right] \\
&= \int_0^s f(r, X_r^{t,x})\mathrm{d}r + \mathbb{E}\left[g(X_T^{t,x}) + \int_0^T f(r, X_r^{t,x})\mathrm{d}r \mid \mathfrak{F}_s\right].
\end{aligned} \tag{8}$$

Now, by the Markov property of $(s, X_s^{t,x} - X_t^{t,x}) = (s, X_s^{t,x} - x)$ for all $s \ge t$, we can write $Y_s^{t,x} = u(s, X_s^{t,x})$ where

$$u(s, y) = \mathbb{E}\left[g(X_T^{s,y}) + \int_s^T f(r, X_r^{s,y})\mathrm{d}r\right].$$

The regularity of $u$ can be checked similarly as in Proposition 2.5 in [19].
Define $\mathbb{G} = (\mathcal{G}_s)_{s \in [0,T]}$ the filtration generated by the deterministic functions $\int_t^T \mathbb{E}\psi(r, X_r^{t,y})\mathrm{d}r$ where $\psi$ is a continuous $\mathbb{R}^q$-valued function. Thus, for any $\mathbb{G}$-measurable $f$ and $g$ such that

$$\mathbb{E}|g(X_T^{t,x})|^2 + \int_0^T \mathbb{E}|f(r, X_r^{t,x})|^2\mathrm{d}r < \infty.$$

Notice that we do not change the filtration here, we have just introduced the appropriate filtration to guarantee the measurability of the deterministic function $u$.

The process $(Y_s^{t,x})_{s \in [0,T]}$ admits a càdlàg version given by $Y_s^{t,x} = u(s, X_s^{t,x})$ thanks to the decomposition (8) as the sum of an absolutely continuous process and a martingale which can be chosen to be càdlàg.

Obviously, the stochastic process $(\widetilde{Y}_s)_{s \in [t,T]}$

$$\widetilde{Y}_s := \int_t^s f(r, X_r^{t,x}) dr + Y_s^{t,x} = \mathbb{E}\left[ g(X_T^{t,x}) + \int_t^T f(r, X_r^{t,x}) dr \mid \mathfrak{F}_s \right], \tag{9}$$

is an additive square-integrable martingale and therefore, by Lemma 4.1. [3] p. 45, or by Lemma 2, with $X$ starting at $x$ at time $t$, it admits the following representation:

$$\widetilde{Y}_s = \int_t^s v(r, X_r^{t,x}) \, dW_r + \int_t^s \int_E \theta(r, X_{r-}^{t,x}, e) \widetilde{N}(dr, de),$$

where $v(r, x) \in \mathbb{R}^{q \times q}$ and $\theta(r, x, \cdot) \in \mathcal{L}_\nu^{2,q}$ are two measurable functions $[0, T] \times \mathbb{R}^p$. Moreover, for $s = T$, we have

$$\widetilde{Y}_T = \int_t^T v(r, X_r^{t,x}) \, dW_r + \int_t^T \int_E \theta(r, X_{r-}^{t,x}, e) \widetilde{N}(dr, de)$$

and then compute the difference $\widetilde{Y}_T - \widetilde{Y}_s$. On the one hand, we have

$$\widetilde{Y}_T - \widetilde{Y}_s = \int_s^T v(r, X_r^{t,x}) \, dW_r + \int_s^T \int_E \theta(r, X_{r-}^{t,x}, e) \widetilde{N}(dr, de).$$

On the other hand, in view of the equality (9),

$$\begin{aligned} \widetilde{Y}_T - \widetilde{Y}_s &= \int_t^T f(r, X_r^{t,x}) dr + Y_T^{t,x} - \int_t^s f(r, X_r^{t,x}) dr + Y_s^{t,x} \\ &= g(X_T^{t,x}) + \int_s^T f(r, X_r^{t,x}) dr - Y_s^{t,x}, \end{aligned}$$

hence,

$$\begin{aligned} Y_s^{t,x} &= g(X_T^{t,x}) + \int_s^T f(r, X_r^{t,x}) dr \\ &\quad - \int_s^T v(r, X_r^{t,x}) \, dW_r - \int_s^T \int_E \theta(r, X_{r-}^{t,x}, e) \widetilde{N}(dr, de) \\ &= g(X_T^{t,x}) + \int_s^T f(r, X_r^{t,x}) dr \\ &\quad - \int_s^T Z_r^{t,x}(e) \, dW_r - \int_s^T \int_E K_r^{t,x}(e) \widetilde{N}(dr, de). \end{aligned}$$

Now, due to the uniqueness of the solution of Equation (7), we obtain

$$Z_r^{t,x} = v(r, X_r^{t,x}) \text{ and } K_r^{t,x}(e) = \theta(r, X_{r-}^{t,x}, e),$$

that is exactly our BSDEJ.

**Step 2.** In this step, we shall consider the general case where the generator $f$ depends on $r$, $x$, $y$, $z$ and $k(\cdot)$. Let us introduce the following sequence $(Y^{t,x,n}, Z^{t,x,n}, K^{t,x,n}(\cdot))_{n \in \mathbb{N}}$ defined by $Y^{t,x,0} = 0$, $Z^{t,x,0} = 0$ and $K^{t,x,0} = 0$ and

$$\begin{aligned} Y_s^{t,x,n+1} &= g(X_T^{t,x}) + \int_s^T f(r, X_r^{t,x,n}, Y_r^{t,x,n}, Z_r^{t,x,n}, K_r^{t,x,n}(\cdot)) dr \\ &\quad - \int_s^T Z_r^{t,x,n+1} \, dW_r - \int_s^T \int_E K_r^{t,x,n+1}(e) \widetilde{N}(dr, de). \end{aligned}$$

Since we are still under the Lipschitz condition of the generator, one can show exactly as in the proof of the Lemma 2.4 in [5] (see also [23–26]) that $(Y_\cdot^{t,x,n}, Z_\cdot^{t,x,n}, K_\cdot^{t,x,n}(\cdot))_{n \in \mathbb{N}}$ is a Cauchy sequence in the Banach space $\mathbb{M}^2$, and hence,

$$(Y_\cdot^{t,x}, Z_\cdot^{t,x}, K_\cdot^{t,x}(\cdot)) = \lim_{n \to +\infty} (Y_\cdot^{t,x,n}, Z_\cdot^{t,x,n}, K_\cdot^{t,x,n}(\cdot)). \tag{10}$$

From the previous step, for any $r \in [t, T]$, we know that there exist three measurable functions $u^1, v^1$ and $\theta^1$ such that

$$(Y_r^{t,x,1}, Z_r^{t,x,1}, K_r^{t,x,1}(e)) = (u^1(r, X_r^{t,x}), v^1(r, X_r^{t,x}), \theta^1(r, X_r^{t,x}, e)), \ \mathbf{P}\text{-a.s.}$$

We conclude, by recursion, for any $n \in \mathbb{N}$ there exist measurable functions $u^n, v^n$, and $\theta^n$ such that $\mathbf{P}$-a.s. $\forall \, r \in [t, T]$

$$(Y_r^{t,x,n}, Z_r^{t,x,n}, K_r^{t,x,n}(e)) = (u^n(r, X_r^{t,x}), v^n(r, X_r^{t,x}), \theta^n(r, X_r^{t,x}, e)). \tag{11}$$

Notice that theses representations have been studied in the literature for smooth coefficients by Barles et al. [19], (Bouchard and Elie [27] [Section 4]) and Delong [24].

Set

$$u(r, X_r^{t,x}) = \lim_{n \to +\infty} \sup u^n(r, X_r^{t,x}), \ v(r, X_r^{t,x}) = \lim_{n \to +\infty} \sup v^n(r, X_r^{t,x}),$$

and

$$\theta(r, X_r^{t,x}, e) = \lim_{n \to +\infty} \sup \theta^n(r, X_r^{t,x}, e).$$

Then, by invoking (10) and (11), it follows that $\mathbf{P}$-a.s. $\forall \, r \in [t, T]$

$$u(r, X_r^{t,x}) = \lim_{n \to +\infty} \sup u^n(r, X_r^{t,x}) = \lim_{n \to +\infty} Y_r^{t,x,n} = Y_r^{t,x}.$$

the same convergence holds true for $v$ and $\theta$. Finally, the linear growth condition on $u$ is a simple consequence of the previous representation in step 2 and Lemma 3. This completes the proof. $\quad \square$

## 4. Markovian BSDEJs with Continuous Generators

Our aim in this section is to handle the existence problem for multidimensional Markovian BSDE driven by both $q$-dimensional Brownian motion and compensated Poisson random measure on $E$. We first study the case when the BSDEJ generator is only continuous with respect to the state variable along with the Brownian component and Lipschitz in the jump component. The idea of the proof is to approximate the BSDEJ under consideration by a suitable sequence of, BSDEJs having globally Lipschitz coefficients that guarantee the existence and uniqueness of solution and then obtain the existence result of the original equation by using limit arguments in appropriate spaces. The drawback to relaxing the Lipschitz condition on $k(\cdot)$ is that it belongs to the functional space $\mathcal{L}_\nu^{2,q}$ and thus, the approximating technique does not work in this situation. However, if we allow the generator $f$ to depend on $\int_E k(e)\nu(\mathrm{d}e)$ rather than $k(\cdot)$, as a particular case, we can prove an existence result in the case where $f$ is also continuous in $k$. Finally, due to the lack of the comparison principle between solutions of multidimensional Markovian BSDE, the technique used in [6] cannot be applied in our situation. As the trade-off, we shall use the relationship between the processes $X_\cdot^{t,x}$ and $(Y_\cdot^{t,x}, Z_\cdot^{t,x}, K_\cdot^{t,x}(\cdot))$ established in Theorem 1 and the $L^2$-domination technique to be defined below.

*4.1. Partially Continuous Case*

Let us consider the BSDEJ of the following type for all $s \in [0, T]$

$$
\begin{aligned}
Y_s^{0,x_0} \;=\; & g(X_T^{0,x_0}) + \int_s^T f(r, X_r^{0,x_0}, Y_r^{0,x_0}, Z_r^{0,x_0}, K_r^{0,x_0}(\cdot)) \mathrm{d}r \\
& - \int_s^T Z_r^{0,x_0} \, \mathrm{d}W_r - \int_s^T \int_E K_r^{0,x_0}(e) \widetilde{N}(\mathrm{d}r, \mathrm{d}e),
\end{aligned}
\tag{12}
$$

where $X_0^{0,x_0} = x_0 \in \mathbb{R}^p$, $g$ is the same as in BSDEJ (4). Throughout this section, we assume that $f$ satisfies the following assumptions:

$(\mathbf{H}_{4.1})$ The mapping $(y, z) \mapsto f(s, x, y, z, k(\cdot))$ is continuous for any fixed

$$
(s, x, k(\cdot)) \in [0, T] \times \mathbb{R}^p \times \mathcal{L}_\nu^{2,q}.
$$

$(\mathbf{H}_{4.2})$ For any $(t, x, y, z) \in [0, T] \times \mathbb{R}^p \times \mathbb{R}^q \times \mathbb{R}^{q \times q}$ and $k, k' \in \mathcal{L}_\nu^{2,q}$

$$
\left| f(t, x, y, z, k(\cdot)) - f(t, x, y, z, k'(\cdot)) \right| \leq C \left\| (k - k')(\cdot) \right\|_{q,\nu}.
$$

Before we state and prove the main results of this section, let us first recall the precise definition of the $L^2$-domination condition, as given in Hamadène [14].

**Definition 2.** *($L^2$-domination condition) For a given $t \in [0, T]$, a family of probability measures $\{\mu_1(s, \mathrm{d}x),\ s \in [t, T]\}$ defined on $\mathbb{R}^p$ is said to be $L^2$-dominated by another family of probability measures $\{\mu_0(s, \mathrm{d}x),\ s \in [t, T]\}$, if for any $\varepsilon \in (0, T - t]$, there exists an application $\phi_t : [t + \varepsilon, T] \times \mathbb{R}^p \longrightarrow \mathbb{R}_+$ such that*
*(i) $\forall N \geq 1,\ \phi_t \in L^2([t + \varepsilon, T] \times [-N, N]^p;\ \mu_0(s, \mathrm{d}x)\mathrm{d}s)$.*
*(ii) $\mu_1(s, \mathrm{d}x)\mathrm{d}s = \phi_t(s, x)\mu_0(s, \mathrm{d}x)\mathrm{d}s$ on $[t + \varepsilon, T] \times \mathbb{R}^p$.*

Let $x_0 \in \mathbb{R}^p$, $(t, x) \in [0, T] \times \mathbb{R}^p$, $s \in [t, T]$ and $\mu(t, x; s, \mathrm{d}y)$ the law of our Markov process $(X_s^{t,x})_{t \leq s \leq T}$, defined for each $A \in \mathcal{B}(\mathbb{R}^p)$, by $\mu(t, x; s, A) = \mathbf{P}(X_s^{t,x} \in A)$.

We further assume the following assumption:

$(\mathbf{H}_{4.3})$ For each $t \geq 0$ and for each $x \in \mathbb{R}^p$ the family $\{\mu(t, x; s, \mathrm{d}y),\ s \in [t, T]\}$ is $L^2$-dominated by $\{\mu(0, x_0; s, \mathrm{d}y),\ s \in [t, T]\}$.

**Lemma 4.** *Let $f$ satisfy $(\mathbf{H}_{3.1})$, $(\mathbf{H}_{3.2})$, $(\mathbf{H}_{4.1})$, and $(\mathbf{H}_{4.2})$. Then, there exists a sequence of functions $(f_n)_{n \geq 1}$ such that:*

**(a)** $\sup_{t,x} |f_n(t, x, y, z, k(\cdot)) - f_n(t, x, y', z', k'(\cdot))|$
$\leq C \Big( |y - y'| + |z - z'| + \|(k - k')(\cdot)\|_{q,\nu} \Big)$, *for some positive constant $C$;*

**(b)** $|f_n(t, x, y, z, k(\cdot))| \leq C(1 + |x| + |y| + |z| + \|k(\cdot)\|_{q,\nu})$, *for all $(t, x, y, z, k(\cdot)) \in [0, T] \times \mathbb{R}^p \times \mathbb{R}^q \times \mathbb{R}^{q \times q} \times \mathcal{L}_\nu^{2,q}$;*

**(c)** *For all $(t, x, y, z, k(\cdot)) \in [0, T] \times \mathbb{R}^p \times \mathbb{R}^q \times \mathbb{R}^{q \times q} \times \mathcal{L}_\nu^{2,q}$ and $n \in \mathbb{N}$, there exists positive a constant $C$ such that $|f_n(t, x, y, z, k(\cdot))| \leq C(1 + |x|)$;*

**(d)** *For any $(t, x, k(\cdot)) \in [0, T] \times \mathbb{R}^p \times \mathcal{L}_\nu^{2,q}$, and for any compact subset $S \subset \mathbb{R}^q \times \mathbb{R}^{q \times q}$*

$$
\sup_{(y,z) \in S} |f_n(t, x, y, z, k(\cdot)) - f(t, x, y, z, k(\cdot))| \longrightarrow 0 \text{ as } n \to +\infty.
$$

**Proof.** Let $\psi$ be an element of $\mathcal{C}^\infty(\mathbb{R}^q \times \mathbb{R}^{q \times q}, \mathbb{R})$ with compact support and satisfy

$$
\int_{\mathbb{R}^{q+q \times d}} \psi(\overrightarrow{u}) \mathrm{d}\overrightarrow{u} = 1,
$$

where $\overrightarrow{u} = (y, z) \in \mathbb{R}^{q+q \times d}$. We define

$$f(t, x, (\cdot)) * \psi(n(\cdot))(\overrightarrow{u}) = \int_{\mathbb{R}^{q+q \times d}} f(t, x, \overrightarrow{v}) \psi(n(\overrightarrow{u} - \overrightarrow{v})) \mathrm{d}\overrightarrow{v}$$

and $\bar{\psi} \in \mathcal{C}^{\infty}(\mathbb{R}^q \times \mathbb{R}^{q \times q}, \mathbb{R})$ such that

$$\bar{\psi}(\overrightarrow{u}) = \left\{ \begin{array}{ll} 1, & |\overrightarrow{u}|^2 \leq 1, \\ 0, & |\overrightarrow{u}|^2 \geq 2. \end{array} \right.$$

Obviously, the sequence of the measurable functions $\{f_n, n \geq 1\}$, such that

$$f_n(t, x, \overrightarrow{u}) = n^2 \bar{\psi}(\frac{\overrightarrow{u}}{n})(f(t, x, (\cdot)) * \psi(n(\cdot)))(\overrightarrow{u}),$$

satisfies all the assertions of Lemma 4. □

Next, we state and prove the first main result in this section. The following theorem extends a part from the paper [14], from BSDEs driven by Brownian motion to BSDEs with jumps. However, our generator depends also on the state variable $y$ which is not covered in [14].

**Theorem 2.** *Assume that* $(\mathbf{H_{3.1}})$–$(\mathbf{H_{3.2}})$ *and* $(\mathbf{H_{4.1}})$–$(\mathbf{H_{4.3}})$ *are in force. Then, there exists a triple of processes* $(Y_\cdot, Z_\cdot, K_\cdot(\cdot))$ *belonging to* $\mathbb{M}^2$ *that solves the BSDEJ* (12).

**Proof.** We first define the following family of approximating BSDEJs obtained by replacing the generator $f$ in BSDEJ (4) by $f_n$ defined in Lemma 4.

$$\begin{aligned} Y_s^{t,x;n} &= g(X_T^{t,x}) + \int_s^T f_n(r, X_r^{t,x}, Y_r^{t,x;n}, Z_r^{t,x;n}, K_r^{t,x;n}(\cdot)) \mathrm{d}r \\ &\quad - \int_s^T Z_r^{t,x;n} \, \mathrm{d}W_r - \int_s^T \int_E K_r^{t,x;n}(e) \widetilde{N}(\mathrm{d}r, \mathrm{d}e). \end{aligned} \qquad (13)$$

On the one hand, since for each $n \geq 1$, $f_n$ is uniformly Lipschitz with respect to $(y, z, k(\cdot))$, so Lemma 2.4 in [5] or [23–26] shows that there exists a unique solution

$$(Y_\cdot^{t,x;n}, Z_\cdot^{t,x;n}, K_\cdot^{t,x;n}(\cdot))_{n \geq 1} \in \mathbb{M}_S^2,$$

which solves BSDEJ (13).

On the other hand, since $f_n$ satisfies the property (**c**) in Lemma 4, Theorem 1 yields that, there exist three sequences of deterministic measurable functions $u^n : [0, T] \times \mathbb{R}^p \longrightarrow \mathbb{R}^q$, $v^n : [0, T] \times \mathbb{R}^p \longrightarrow \mathbb{R}^{q \times q}$ and $\theta^n : [0, T] \times \mathbb{R}^p \longrightarrow \mathcal{L}_v^{2,q}$ such that

$$Y_s^{t,x;n} = u^n(s, X_s^{t,x}), \ Z_s^{t,x;n} = v^n(s, X_s^{t,x}) \text{ and } K_s^{t,x;n}(e) = \theta^n(s, X_{s-}^{t,x}, e).$$

Furthermore, we have the following deterministic expression for the function $u^n$ such that for $n \geq 1$,

$$u^n(t, x) = \mathbb{E}\left[ g(X_T^{t,x}) + \int_t^T F_n(s, X_s^{t,x}) \mathrm{d}s \right], \ \forall \, (t, x) \in [0, T] \times \mathbb{R}^p, \qquad (14)$$

where

$$F_n(t, x) = f_n(t, x, u^n(t, x), v^n(t, x), \theta^n(t, x, \cdot)).$$

Hence, keeping in mind the property **(b)**, as in Lemma 3, one can show that there exists a constant $C > 0$ (independent from $n$) such that for any $n \geq 1$ and $s \in [t, T]$,

$$\mathbb{E}\left[\left|u^n(s, X_s^{t,x})\right|^2 + \int_0^T \left(\left|Z_r^{t,x;n}\right|^2 + \left\|K_r^{t,x;n}(\cdot)\right\|_{q,\nu}^2\right)dr\right] \leq C(1 + |x|^2).$$

In particular, due to the fact that $X_t^{t,x} = x$, we obtain

$$|u^n(t, x)|^2 \leq C(1 + |x|^2) \ \forall \, n \geq 1 \text{ and } t \in [0, T],$$

consequently $|u^n(t, x)| \leq C(1 + |x|) \ \forall \, n \geq 1$ and $t \in [0, T]$, and thus, for any $s \in [0, T]$ and $n \geq 1$,

$$|Y_s^n| = \left|u^n(s, X_s^{0,x_0})\right| \quad \leq \quad C(1 + |X_s^{0,x_0}|), d\mathbf{P}\text{-a.s.} \tag{15}$$

The remainder of the proof will be broken down into the following three steps.
**Step 1.** In this step, we will prove, for each $(t, x) \in [0, T] \times \mathbb{R}^p$, that $(u^n(t, x))_{n \geq 1}$ has a convergent subsequence in $\mathbb{R}^q$. On one hand, since $f_n$ satisfies the property **(b)**, the same technique as in the proof of Lemma 3, yields

$$\sup_{n \geq 1} \mathbb{E}\left[\left|Y_s^{0,x_0;n}\right|^2 + \int_0^T \left(\left|Z_r^{0,x_0;n}\right|^2 + \left\|K_r^{0,x_0;n}(\cdot)\right\|_{q,\nu}^2\right)dr\right] \quad \leq \quad C, \ \forall \, s \leq T. \tag{16}$$

We now apply the property **(b)** with Assumption $(\mathbf{H}_{4.3})$, $(\mathbf{H}_{3.1})$, and the estimate (16) to show that there exists a positive constant $C$ such that, for any $n \geq 1$,

$$\int_0^T \int_{\mathbb{R}^p} |F_n(s, y)|^2 \mu(0, x_0; s, dy)ds = \mathbb{E}\int_0^T \left|F_n(s, X_s^{0,x_0})\right|^2 ds \quad \leq \quad C. \tag{17}$$

Therefore, there exists a subsequence $\{n_j\}$, (which is still labeled by $\{n\}$), and $\mathcal{B}([0, T]) \otimes \mathcal{B}(\mathbb{R}^p)$-measurable deterministic function $F(s, x)$ such that

$$F_n \quad \longrightarrow \quad F \text{ weakly in } L^2([0, T] \times \mathbb{R}^p; \mu(0, x_0; s, dxds)). \tag{18}$$

On the other hand, let $(t, x)$ be fixed, $\varepsilon > 0$, $N$, $n$, and $m \geq 1$ be integers. Then, from (14), we have

$$\begin{aligned}
|u^n(t, x) - u^m(t, x)| &= \left|\mathbb{E}\left[\int_t^T (F_n(s, X_s^{t,x}) - F_m(s, X_s^{t,x}))ds\right]\right| \\
&\leq \mathbb{E}\left[\int_t^{t+\varepsilon} \left|F_n(s, X_s^{t,x}) - F_m(s, X_s^{t,x})\right|ds\right] \\
&\quad + \left|\mathbb{E}\left[\int_{t+\varepsilon}^T (F_n(s, X_s^{t,x}) - F_m(s, X_s^{t,x}))\mathbb{1}_{\{|X_s^{t,x}| \leq N\}}ds\right]\right| \\
&\quad + \mathbb{E}\left[\int_{t+\varepsilon}^T \left|F_n(s, X_s^{t,x}) - F_m(s, X_s^{t,x})\right|\mathbb{1}_{\{|X_s^{t,x}| > N\}}ds\right] \\
&= I_1^{n,m,\varepsilon} + \left|I_2^{n,m,\varepsilon}\right| + I_3^{n,m,\varepsilon}. \tag{19}
\end{aligned}$$

We first estimate $I_1^{n,m,\varepsilon}$. According to the Schwarz inequality and (17), we obtain

$$I_1^{n,m,\varepsilon} \leq \varepsilon^{\frac{1}{2}}\left\{\mathbb{E}\left[\int_0^T \left|F_n(s, X_s^{t,x}) - F_m(s, X_s^{t,x})\right|^2 ds\right]\right\}^{\frac{1}{2}} \leq C\sqrt{\varepsilon}.$$

Next, the $L^2$-domination property implies

$$
\begin{aligned}
I_2^{n,m,\varepsilon} &= \int_{\mathbb{R}^p} \int_{t+\varepsilon}^T (F_n(s,y) - F_m(s,y)) \mathbb{1}_{\{|y| \le N\}} \mu(t,x;s,\mathrm{d}y) \mathrm{d}s \\
&= \int_{\mathbb{R}^p} \int_{t+\varepsilon}^T (F_n(s,y) - F_m(s,y)) \mathbb{1}_{\{|y| \le N\}} \phi_{t,x}(s,y) \mu(0,x_0;s,\mathrm{d}y) \mathrm{d}s.
\end{aligned}
$$

Since $\phi_{t,x}(s,y) \in L^2([t+\varepsilon, T] \times [-N,N]^p; \mu(0,x_0;s,\mathrm{d}y)\mathrm{d}s)$, for $k \ge 1$, it follows from (18) that for any $t \ge 0$, $\mu(0,x_0;s,\mathrm{d}y)$-almost every $x \in \mathbb{R}^p$, we have

$$
\mathbb{E}\left[ \int_{t+\varepsilon}^T (F_n(s, X_s^{t,x}) - F_m(s, X_s^{t,x})) \mathbb{1}_{\{|X_s^{t,x}| \le N\}} \mathrm{d}s \right] \longrightarrow 0 \text{ as } n, \, m \longrightarrow \infty.
$$

Finally,

$$
\begin{aligned}
I_3^{n,m,\varepsilon} &\le \left\{ \mathbb{E}\left[ \int_{t+\varepsilon}^T \mathbb{1}_{\{|X_s^{t,x}| > N\}} \mathrm{d}s \right] \right\}^{\frac{1}{2}} \left\{ \mathbb{E}\left[ \int_{t+\varepsilon}^T \left| F_n(s, X_s^{t,x}) - F_m(s, X_s^{t,x}) \right|^2 \mathrm{d}s \right] \right\}^{\frac{1}{2}} \\
&\le \frac{C}{\sqrt{N}}.
\end{aligned}
$$

Therefore, by letting $N$ and $(m,n)$ tend to infinity successively, the sequence $(u^n(t,x))_{n \ge 1}$ has a convergence subsequence in $\mathbb{R}^q$ with limit $u(t,x)$ for any $t \ge 0$ and every $x \in \mathbb{R}^p$.

**Step 2.** We are going to show the existence of a subsequence still denoted

$$
(Y_\cdot^{0,x_0;n}, Z_\cdot^{0,x_0;n}, K_\cdot^{0,x_0;n})_{n \ge 1},
$$

which converges in $\mathbb{M}^2$ to $(Y_\cdot, Z_\cdot, K_\cdot(\cdot))$ solution of the BSDEJ (12).

From step 1, there exists a measurable function $u$ on $[0,T] \times \mathbb{R}^p$, such that for any $t \in [0,T]$,

$$
\lim_{n \to +\infty} Y_t^{0,x_0;n} = u(t, X_t^{0,x_0}), \quad \textbf{P}\text{-a.s.}
$$

Considering (15), and using Lebesgue's dominated convergence Theorem, the sequence $(Y_t^{0,x_0;n})_{n \ge 1}$ converges to $Y_t^{0,x_0} := u(t, X_t^{0,x_0})$ in $\mathcal{M}_{\mathfrak{F}}^2(0, T, \mathbb{R}^q)$, that is,

$$
\mathbb{E}\left[ \int_0^T \left| Y_t^{0,x_0;n} - Y_t^{0,x_0} \right|^2 \mathrm{d}t \right] \longrightarrow 0. \tag{20}
$$

Next, we will show the convergence of $(Z_\cdot^{0,x_0;n})_{n \ge 1}$ and $(K_\cdot^{0,x_0;n})_{n \ge 1}$ respectively in $\mathcal{M}_{\mathfrak{F}}^2(0, T, \mathbb{R}^{q \times q})$ and $\mathcal{M}_{\mathfrak{F}}^2([0,T] \times E, \mathbb{R}^q, \mathrm{d}t \nu(\mathrm{d}e))$ as $n \to +\infty$. For the sake of convenience, we omit the subscript $(0, x_0)$.

To simplify the notations, for any $n, m \ge 1$, and $s \le T$, we set:

$$
\bar{Y}_s^{n,m} := Y_s^n - Y_s^m, \quad \bar{Z}_s^{n,m} := Z_s^n - Z_s^m \text{ and } \bar{K}_s^{n,m}(\cdot) := K_s^n(\cdot) - K_s^m(\cdot)
$$

and

$$
\bar{f}^{n,m}(s) := f_n(s, X_s^{0,x_0}, Y_{s-}^n, Z_s^n, K_s^n(\cdot)) - f_m(s, X_s^{0,x_0}, Y_{s-}^m, Z_s^m, K_s^m(\cdot)).
$$

Itô's formula applied to $\left| \bar{Y}_s^{n,m} \right|^2$ leads to

$$
\begin{aligned}
&\left| \bar{Y}_s^{n,m} \right|^2 + \int_s^T |\bar{Z}_r^{n,m}|^2 \mathrm{d}r + \int_s^T \|\bar{K}_r^{n,m}(\cdot)\|_{q,\nu} \mathrm{d}r \\
&= 2 \int_s^T \bar{Y}_r^{n,m} \, \bar{f}^{n,m}(r) \mathrm{d}r - (M_T^{n,m} - M_s^{n,m}) - (N_T^{n,m} - N_s^{n,m}), \tag{21}
\end{aligned}
$$

where

$$
M_s^{n,m} = 2 \int_0^s \bar{Y}_r^{n,m} \, \bar{Z}_r^{n,m} \, \mathrm{d}W_r - 2 \int_0^s \int_E \bar{Y}_r^{n,m} \, \bar{K}_r^{n,m}(e) \widetilde{N}(\mathrm{d}r, \mathrm{d}e),
$$

and

$$N_s^{n,m} = \int_0^s \int_E |\bar{K}_r^{n,m}(e)|^2) \widetilde{N}(dr, de).$$

are real-valued martingales. Hence, according to property **(b)**, the Assumption $(\mathbf{H}_{3.1})$ and the estimate (16), we obtain by taking the expectation in (21)

$$\int_0^T \mathbb{E}\left[|\bar{Z}_r^{n,m}|^2 + \|\bar{K}_r^{n,m}(\cdot)\|_{q,\nu}^2\right] dr \le C\left[\int_0^T \mathbb{E}\left[|\bar{Y}_r^{n,m}|^2\right] dr\right].$$

Thanks to (20), it follows that $((Z_\cdot^n)_{n\ge 1}, (K_\cdot^n(\cdot))_{n\ge 1}))$ converges to some $(Z_\cdot, K_\cdot(\cdot))$ in $\mathcal{M}_{\mathfrak{F}}^2(0, T, \mathbb{R}^{q\times q}) \times \mathcal{M}_{\mathfrak{F}}^2([0,T] \times E, \mathbb{R}^q, dt\nu(de)))$. Finally, we have proved that for a subsequence $n_j$,

$$(Y_\cdot^{n_j}, Z_\cdot^{n_j}, K_\cdot^{n_j}(\cdot))_{j\ge 1} \longrightarrow (Y_\cdot, Z_\cdot, K_\cdot(\cdot)) \text{ in } \mathbb{M}^2. \tag{22}$$

**Step 3.** In this step, we will verify that the limits of the subsequences are exactly the solutions to BSDEJ (12). It remains to prove that $f_n(t, X_t^{0,x_0}, Y_t^n, Z_t^n, K_t^n(\cdot))$ converges to $f(t, X_t^{0,x_0}, Y_t, Z_t, K_t(\cdot)) \, dt \otimes d\mathbf{P}$. For $N \ge 1$, we define

$$A_N := \{(r, \omega) : |Y_r^n| + |Z_r^n| \le N\}, \quad \bar{A}_N := \Omega \setminus A_N, \tag{23}$$

then, we have

$$\mathbb{E}\left[\int_0^T \left|f_n(r, X_r^{0,x_0}, Y_r^n, Z_r^n, K_r^n(\cdot)) - f(r, X_r^{0,x_0}, Y_r, Z_r, K_r(\cdot))\right| dr\right]$$

$$\le \mathbb{E}\left[\int_0^T \left|f_n(r, X_r^{0,x_0}, Y_r^n, Z_r^n, K_r^n(\cdot)) - f_n(r, X_r^{0,x_0}, Y_r^n, Z_r^n, K_r(\cdot))\right| dr\right]$$

$$+ \mathbb{E}\left[\int_0^T \left|(f_n - f)(r, X_r^{0,x_0}, Y_r^n, Z_r^n, K_r(\cdot))\right| \mathbb{1}_{A_N} dr\right],$$

$$+ \mathbb{E}\left[\int_0^T \left|(f_n - f)(r, X_r^{0,x_0}, Y_r^n, Z_r^n, K_r(\cdot))\right| \mathbb{1}_{\bar{A}_N} dr\right],$$

$$+ \mathbb{E}\left[\int_0^T \left|f(r, X_r^{0,x_0}, Y_r^n, Z_r^n, K_r(\cdot)) - f(r, X_r^{0,x_0}, Y_r, Z_r, K_r(\cdot))\right| dr\right],$$

$$= I_1^n + I_2^n + I_3^n + I_4^n.$$

Thanks to **(a)** in Lemma 4 and (22) $I_1^n$ converges to 0. Again, according to (22) and the continuity of $f$ with respect to $(y, z, k(\cdot))$, it follows, using Assumptions $(\mathbf{H}_{3.1})$, $(\mathbf{H}_{3.2})$ and Lebesgue's dominated theorem, that $I_4^n$ converges to 0 as $n$ tends toward infinity. Moreover, since $f_n$ satisfies the property **(b)** in Lemma 4, $f$ satisfies $(\mathbf{H}_{3.1})$, $(\mathbf{H}_{3.2})$, and the estimate (16), a simple computation shows that there is a positive constant $C$ such that $I_3^n \le CN^{-\frac{1}{2}}$. Now, we return to estimate the second term $I_2^n$. The linear growth condition **(b)** together with $(\mathbf{H}_{3.1})$ and $(\mathbf{H}_{3.2})$ imply that:

$$\left|(f_n - f)(r, X_r^{0,x_0}, Y_r^n, Z_r^n, K_r(\cdot))\right| \mathbb{1}_{A_N} \le 2C(1 + N + \left|X_r^{0,x_0}\right|).$$

On the other hand, it is easy to see that

$$\left|(f_n - f)(r, X_r^{0,x_0}, Y_r^n, Z_r^n, K_r(\cdot))\right| \mathbb{1}_{A_N}$$

$$\le \sup_{\{(y,z), |y| + |z| \le N\}} \left|(f_n - f)(r, X_r^{0,x_0}, y, z, K_r(\cdot))\right|.$$

Then, from the property **(d)** in Lemma 4, we conclude that the second term of the last inequality converges to 0. Finally, Lebesgue's dominated convergence theorem asserts that $I_2^n$ converges also to 0 in $L^1([0,T] \times \Omega, dt \otimes d\mathbb{P})$.

Eventually, we find, by sending $n$ and $N$ to infinity, the converges of sequence

$$(f_n(t, X_t^{0,x_0}, Y_t^n, Z_t^n, K_t^n(\cdot)))_{0 \le t \le T})_{n \ge 1}$$

to

$$(f(t, X_t^{0,x_0}, Y_t, Z_t, K_t(\cdot)))_{0 \le t \le T}$$

in $L^1([0, T] \times \Omega, dt \otimes d\mathbb{P})$, and then $F(t, X_t^{0,x_0}) = f(t, X_t^{0,x_0}, Y_t, Z_t, K_t(\cdot))$, $dt \otimes d\mathbf{P}$-a.s. It follows clearly that $(Y_\cdot, Z_\cdot, K_\cdot(\cdot))$ solves Equation (12). □

To obtain the convergence in $\mathbb{M}_\mathcal{S}^2$, we add the following assumption on the generator $f$:

**($\mathbf{H}_{4.4}$)** $f$ : $[0, T] \times \Omega \times \mathbb{R}^p \times \mathbb{R}^q \times \mathbb{R}^{q \times q} \times \mathcal{L}_\nu^{2,q} \longrightarrow \mathbb{R}^q$ is measurable and for any $(t, x, y, z, k(\cdot)) \in [0, T] \times \mathbb{R}^p \times \mathbb{R}^q \times \mathbb{R}^{q \times q} \times \mathcal{L}_\nu^{2,q}$, there exists a constant $C > 0$ and $0 \le \beta < 1$ such that

$$|f(t, x, y, z, k(\cdot))| \le C(1 + |x| + |y| + |z| + \|k(\cdot)\|_{q,\nu})^\beta.$$

**Corollary 1.** *Assume that* **($\mathbf{H}_{3.1}$)** *and* **($\mathbf{H}_{4.1}$)**−**($\mathbf{H}_{4.4}$)** *are in force. Then, there exists a triple of processes* $(Y_\cdot, Z_\cdot, K_\cdot(\cdot))$ *belonging to* $\mathbb{M}_\mathcal{S}^2$ *that solves the BSDEJ* (12).

**Proof.** By using the inequality $|x|^\beta \le 1 + |x|$, it is easy to check that the sub-linear growth condition **($\mathbf{H}_{4.4}$)** implies the linear growth condition **($\mathbf{H}_{3.2}$)**, thus the above theorem confirms that there exists a triple $(Y_\cdot, Z_\cdot, K_\cdot(\cdot))$ solution to the BSDEJ (12) which belongs to $\mathbb{M}^2$. It remains to prove that the sequence $(Y_s^n)_{n \ge 1} = (u^n(s, X_s^{(0,x_0)}))_{n \ge 1}$ defined in the above proof converges to $Y_\cdot$ in $\mathcal{S}_\mathfrak{F}^2(0, T, \mathbb{R}^q)$. From (12) and (13), squaring both sides of $(Y_r^n - Y_r)$, taking the supremum, the conditional expectation, using BDG inequality, we obtain

$$\mathbb{E}\left[\sup_{0 \le r \le T} |Y_r^n - Y_r|^2\right] \le C\left[\mathbb{E}\int_0^T \left|(f_n - f)(r, X_r^{0,x_0}, Y_r^n, Z_r^n, K_r(\cdot))\right|^2 \mathbb{1}_{A_N} dr \right. \tag{24}$$

$$+ \mathbb{E}\int_0^T \left|(f_n - f)(r, X_r^{0,x_0}, Y_r^n, Z_r^n, K_r(\cdot))\right|^2 \mathbb{1}_{\bar{A}_N} dr$$

$$+ \mathbb{E}\int_0^T \left|f(r, X_r^{0,x_0}, Y_r^n, Z_r^n, K_r(\cdot)) - f(r, X_r^{0,x_0}, Y_r, Z_r, K_r(\cdot))\right|^2 dr$$

$$\left. + \mathbb{E}\int_0^T |Z_r^n - Z_r|^2 dr + \mathbb{E}\int_0^T \|(K_r^n - K_r)(\cdot)\|_{q,\nu}^2 dr\right],$$

where $A_N$ and $\bar{A}_N$ are defined by (23).

Since $(Z_\cdot^n)_{n \ge 1}$ (respectively $(K_\cdot^n(\cdot))_{n \ge 1}$) converges in $\mathcal{L}_\mathfrak{F}^2(0, T, \mathbb{R}^{q \times q})$ (respectively $\mathcal{M}_\mathfrak{F}^2([0, T] \times E, \mathbb{R}^q, dt\nu(de))$) to $Z_\cdot$ (respectively, $K_\cdot(\cdot)$), the fourth and fifth terms on the right-hand side of the above inequality tends to 0 as $n$ goes towards infinity. Then, by using similar arguments to estimate $I_2^n$ (respectively, $I_4^n$) in the previous step, one can prove that the first (respectively, the third) term also tends to 0 as $n$ tends to infinity.

Next, we proceed to estimate the second term in inequality (24). Since $f_n$ satisfies the property **(b)** in Lemma 4 and $f$ satisfies **($\mathbf{H}_{4.4}$)**, we have

$$\mathbb{E}\left[\int_0^T \left|(f_n - f)(r, X_r^{0,x_0}, Y_r^n, Z_r^n, K_r(\cdot))\right|^2 \mathbb{1}_{\bar{A}_N} dr\right]$$

$$\le C\mathbb{E}\left[\int_0^T (1 + \left|X_r^{0,x_0}\right| + |Y_r^n| + |Z_r^n| + \|K_r(\cdot)\|_{q,\nu})^{2\beta} \mathbb{1}_{\bar{A}_N} dr\right].$$

Thanks to Hölder's inequality applied with $p = \frac{1}{\beta}$ and $q = \frac{1}{1-\beta}$, along with **($\mathbf{H}_{3.1}$)** and the estimate (16), one can show that

$$\mathbb{E}\left[\int_0^T \left|(f_n - f)(r, X_r^{0,x_0}, Y_r^n, Z_r^n, K_r(\cdot))\right|^2 \mathbb{1}_{\bar{A}_N} dr\right] \leq C(T\mathbf{P}(\bar{A}_N))^{1-\beta}.$$

Chebyshev's inequality yields that there is a positive constant $C$ such that

$$\mathbb{E}\left[\int_0^T \left|(f_n - f)(r, X_r^{0,x_0}, Y_r^n, Z_r^n, K_r^n(\cdot))\right|^2 \mathbb{1}_{\bar{A}_N} dr\right] \leq CN^{\beta-1}.$$

By letting $N$ tend to infinity, the previous inequality tends to 0 as $n$ goes to infinity. Consequently, $(Y_\cdot^n)_{n\geq 1}$ converges to $Y_\cdot$ in $\mathcal{S}^2_{\tilde{\mathfrak{F}}}(0, T, \mathbb{R}^q)$. $\square$

**Remark 2.** *Note that all the previous results of this section still hold true if the Markov process $X_\cdot$ is constant with value $x \in \mathbb{R}^p$ on $[0, t]$.*

*4.2. Totally Continuous Case*

So far, we have investigated the existence results for BSDEs driven by a Brownian motion and a compensated Poisson random measure. The case of BSDEJ with continuous generators $y$ and $z$ and Lipschitz continuous on $k(\cdot)$ is considered. We claim that, from a mathematics point of view, it is hard to deal with the general case where the generator $f$ is continuous in $(y, z, k(\cdot))$. Indeed, the difficulty comes from the fact that the process $k(\cdot)$ takes values in the functional space $\mathcal{L}_\nu^{2,q}$ not in $\mathbb{R}^q$. In the remainder of this paper, we try to relax the globally Lipschitz condition on $k(\cdot)$ by considering the following special case

$$\begin{aligned} Y_s^{t,x} &= g(X_T^{t,x}) + \int_s^T f\left(r, X_r^{t,x}, Y_r^{t,x}, Z_r^{t,x}, \int_E K_r^{t,x}(e)\nu(de)\right)dr \\ &\quad - \int_s^T Z_r^{t,x} \, dW_r - \int_s^T \int_E K_r^{t,x}(e)\tilde{N}(dr, de), \end{aligned} \tag{25}$$

where $X_s^{t,x} = x$ for $s \in [0, t]$. For a given measurable function $f$ defined from $[0, T] \times \mathbb{R}^p \times \mathbb{R}^q \times \mathbb{R}^{q \times q} \times \mathbb{R}^q$ into $\mathbb{R}^q$, we consider the three assumptions:

$(\mathbf{H}_{4.5})$ for any $(t, x, y, z, k) \in [0, T] \times \mathbb{R}^p \times \mathbb{R}^q \times \mathbb{R}^{q \times q} \times \mathbb{R}^q$, there exists a constant $C > 0$ such that
$$|f(t, x, y, z, k)| \leq C(1 + |x| + |y| + |z| + |k|).$$

$(\mathbf{H}_{4.6})$ for any $(t, x, y, z, k) \in [0, T] \times \mathbb{R}^p \times \mathbb{R}^q \times \mathbb{R}^{q \times q} \times \mathbb{R}^q$, there exists a constant $C > 0$ and $0 \leq \beta < 1$ such that
$$|f(t, x, y, z, k)| \leq C(1 + |x| + |y| + |z| + |k|)^\beta.$$

$(\mathbf{H}_{4.7})$ the mapping $(y, z, k) \longmapsto f(t, x, y, z, k)$ is continuous for any fixed $(t, x) \in [0, T] \times \mathbb{R}^p$.

**Theorem 3.** *Under $(\mathbf{H}_{3.1})$, $(\mathbf{H}_{4.5})$, and $(\mathbf{H}_{4.7})$ BSDEJ (25) has at least one solution $(Y_\cdot, Z_\cdot, K_\cdot(\cdot))$ which belongs to $\mathbb{M}^2$. Furthermore, if $f$ satisfies $(\mathbf{H}_{3.1})$, $(\mathbf{H}_{4.6})$, and $(\mathbf{H}_{4.7})$, then the solution is in $\mathbb{M}^2_{\mathcal{S}}$.*

**Proof.** For a given $\Psi$, an element of $\mathcal{C}^\infty(\mathbb{R}^q \times \mathbb{R}^{q \times q} \times \mathbb{R}^q, \mathbb{R})$ with a compact support such that
$$\int_{\mathbb{R}^{q+q \times q+q}} \Psi(\overrightarrow{u}) d\overrightarrow{u} = 1,$$
where $\overrightarrow{u} = (y, z, k) \in \mathbb{R}^{q+q \times q+q}$. We define
$$f(t, x, (\cdot)) * \Psi(n(\cdot))(\overrightarrow{u}) = \int_{\mathbb{R}^{q+q \times q+q}} f(t, x, \overrightarrow{v})\Psi(n(\overrightarrow{u} - \overrightarrow{v}))d\overrightarrow{v}$$
and $\bar{\Psi} \in \mathcal{C}^\infty(\mathbb{R}^q \times \mathbb{R}^{q \times q} \times \mathbb{R}^q, \mathbb{R})$ such that

$$\bar{\Psi}(\overrightarrow{u}) = \begin{cases} 1, & |\overrightarrow{u}|^2 \leq 1, \\ 0, & |\overrightarrow{u}|^2 \geq 2. \end{cases}$$

Let $f$ be a function satisfying ($\mathbf{H}_{4.5}$) and ($\mathbf{H}_{4.7}$). The sequence of the measurable functions $\{f_n, n \geq 1\}$, defined by

$$f_n(t, x, \overrightarrow{u}) = n^3 \bar{\Psi}(\frac{\overrightarrow{u}}{n})(f(t, x, (\cdot)) * \Psi(n(\cdot)))(\overrightarrow{u}),$$

satisfies the following proprieties

**(i)** $\sup_{t,x} |f_n(t, x, y, z, k) - f_n(t, x, y', z', k')| \leq C(|y - y'| + |z - z'| + |k - k'|)$ for some positive constant $C$;

**(ii)** $|f_n(t, x, y, z, k)| \leq C(1 + |x| + |y| + |z| + |k|)$, for all $(t, x, y, z, k)$ in the product space $[0, T] \times \mathbb{R}^p \times \mathbb{R}^q \times \mathbb{R}^{q \times q} \times \mathbb{R}^q$;

**(iii)** For all $(t, x, y, z, k) \in [0, T] \times \mathbb{R}^p \times \mathbb{R}^q \times \mathbb{R}^{q \times q} \times \mathbb{R}^q$ and $n \in \mathbb{N}$, there exists positive constant $C$ such that $|f_n(t, x, y, z, k)| \leq C(1 + |x|)$;

**(iv)** For any $(t, x) \in [0, T] \times \mathbb{R}^p$, and for any compact subset $S \subset \mathbb{R}^q \times \mathbb{R}^{q \times q} \times \mathbb{R}^q$,

$$\sup_{(y,z,k) \in S} |f_n(t, x, y, z, k) - f(t, x, y, z, k)| \longrightarrow 0 \text{ as } n \to +\infty.$$

Firstly, to prove the existence, we define the following family of approximating BSDEJs obtained by replacing the generator $f$ in BSDEJ (25) with $f_n$ defined above.

$$\begin{aligned} Y_s^{t,x;n} = {} & g(X_T^{t,x}) + \int_s^T f_n\left(r, X_r^{t,x}, Y_r^{t,x;n}, Z_r^{t,x;n}, \int_E K_r^{t,x;n}(e)\nu(\mathrm{d}e)\right)\mathrm{d}r \\ & - \int_s^T Z_r^{t,x;n}\,\mathrm{d}W_r - \int_s^T \int_E K_r^{t,x;n}(e)\widetilde{N}(\mathrm{d}r, \mathrm{d}e). \end{aligned} \tag{26}$$

By Lemma 2.4 in [5] Equation (26) admits a unique solution denoted

$$(Y_{\cdot}^{t,x;n}, Z_{\cdot}^{t,x;n}, K_{\cdot}^{t,x;n}(\cdot))_{n \geq 1}$$

and belongs to $\mathbb{M}^2$. Taking into account that $f_n$ satisfies the above (iii), Theorem 1 confirms the existence of three sequences of measurable and deterministic functions $u^n : [0, T] \times \mathbb{R}^p \longrightarrow \mathbb{R}^q$, $v^n : [0, T] \times \mathbb{R}^p \longrightarrow \mathbb{R}^{q \times q}$ and $\theta^n : [0, T] \times \mathbb{R}^p \longrightarrow \mathcal{L}_{\nu}^{2,q}$ such that

$$Y_r^{t,x;n} = u^n(r, X_r^{t,x}), \ Z_r^{t,x;n} = v^n(r, X_r^{t,x}) \text{ and } K_r^{t,x;n}(e) = \theta^n(r, X_{r-}^{t,x}, e).$$

Furthermore, we have the following equality

$$u^n(t, x) = \mathbb{E}\left[g(X_T^{t,x}) + \int_t^T F_n(r, X_r^{t,x})\mathrm{d}r\right], \ \forall (t, x) \in [0, T] \times \mathbb{R}^p,$$

where we have denoted by

$$F_n(t, x) = f_n\left(t, x, u^n(t, x), v^n(t, x), \int_E \theta^{(n)}(t, x, e)\nu(\mathrm{d}e)\right).$$

Starting from the sequence defined in (26) and reasoning as in the three steps of the proof of Theorem 2, we can also establish the existence of at least one solution $(Y_{\cdot}^{t,x}, Z_{\cdot}^{t,x}, K_{\cdot}^{t,x}(\cdot))$ to BSDEJ (25) which belongs to $\mathbb{M}^2$ provided that ($\mathbf{H}_{3.1}$), ($\mathbf{H}_{4.5}$), and ($\mathbf{H}_{4.7}$) hold true. Furthermore, using similar arguments in the proof of Corollary 1, one can prove that the solution $(Y_{\cdot}^{t,x}, Z_{\cdot}^{t,x}, K_{\cdot}^{t,x}(\cdot))$ is in fact in $\mathbb{M}_S^2$ whenever ($\mathbf{H}_{3.1}$), ($\mathbf{H}_{4.6}$), and ($\mathbf{H}_{4.7}$) are in force.　□

*4.3. Examples of Markov Processes Satisfying $L^2$-Domination Condition*

Let us give some examples of Markov processes satisfying our assumption $(\mathbf{H}_{4.3})$.

1. Obviously, the Brownian Motion with drift, starting at $x$ at time $t$: $X_s^{t,x} = B_s^{t,x} + as$, where $B_s^{t,x} = x \in \mathbb{R}^p$ for all $s \leq t$ and $B^{t,x}$ is an $\mathbb{R}^p$-valued Brownian motion and $a \in \mathbb{R}^p$, satisfies the $(\mathbf{H}_{4.3})$ since it has a density;

2. Let us now consider a more general Markov process solution to the following SDE

$$X_s^{t,x} = x + \int_t^s b(r, X_r^{t,x}) \mathrm{d}r + \int_t^s \sigma(r, X_r^{t,x}) \mathrm{d}W_r, \tag{27}$$

with $X_s^{t,x} = x$ if $s \leq t$. The functions $b: [0, T] \times \mathbb{R}^p \longrightarrow \mathbb{R}^p$, $\sigma: [0, T] \times \mathbb{R}^p \longrightarrow \mathbb{R}^{p \times q}$, satisfy the following conditions:

   (a) $\sigma$ is Lipschitz with respect to $x$ uniformly in $t$;
   (b) $\sigma$ is invertible and bounded and its inverse is bounded;
   (c) $b$ is Lipschitz with respect to $x$ uniformly in $t$ and of linear growth.
   According to Lemma 4.3 in [14], the law $\mu(t, x; s, \mathrm{d}y)$ of $X_s^{t,x}$ satisfies $(\mathbf{H}_{4.3})$.

3. Let $(W_s^{t,x})_{t \leq s \leq T}$ be an $\mathbb{R}^d$-valued Brownian motion such that $W_s^{t,x} = x$ if $s \leq t$ and $(A_s)_{0 \leq s \leq T}$ an $\frac{\alpha}{2}$-stable subordinator starting at zero, $0 < \alpha < 2$, independent of $W^{t,x}$ for every $\mathbf{P}_x$. Set $X_s^{t,x} = W_{A_r}^{t,x}$ a rotationally invariant $\alpha$-stable process whose generator is the fractional power of order $\frac{\alpha}{2}$ of the negative Laplacian, corresponding to the Riesz potential of order $\alpha$.
   It is well known that $X^{t,x}$ is a Markov process and the law $\mu(t, x; s, \mathrm{d}y)$ of $X_s^{t,x}$ has a transition density $p(t, x; s, y)$ satisfying the following upper and lower estimates

$$c_1(s-t)^{-\frac{d}{\alpha}} \wedge \frac{s-t}{|x-y|^{d+\alpha}} \leq p(t, x; s, y)$$

$$= \quad p(t-s, x-y) \leq c_2(s-t)^{-\frac{d}{\alpha}} \wedge \frac{s-t}{|x-y|^{d+\alpha}},$$

for all $s \geq t$, and $x, y \in \mathbb{R}^d$. Under a simple relation between $\alpha$ and $d$, the law $\mu(t, x; s, \mathrm{d}y)$ of $X_s^{t,x}$ satisfies $(\mathbf{H}_{4.3})$.

4. Let $D$ be an open subset of $\mathbb{R}^d$ and $\tau_D^X = \inf\{s > 0 : X_s \notin D\}$ be the exit time of $X$ from $D$. The process $X$ killed upon exiting $D$ is defined by

$$X_s^D = \left\{ \begin{array}{ll} X_s & \text{if} \quad s < \tau_D^X \\ \varkappa & \text{if} \quad s \geq \tau_D^X, \end{array} \right. = \left\{ \begin{array}{ll} W_{A_s} & \text{if} \quad s < \tau_D^X \\ \varkappa & \text{if} \quad s \geq \tau_D^X, \end{array} \right.$$

where $\varkappa$ is an isolated point. The infinitesimal generator of the Markov process $X^D$ is the Dirichlet fractional Laplacian, $-(-\Delta)^{\frac{\alpha}{2}}|_D$, i.e., the fractional Laplacian with zero exterior conditions.
   It is shown in ([28], Theorem 1.1) that when $D$ is a $\mathcal{C}^{1,1}$ open set in $\mathbb{R}^d$, $d \geq 1$ the heat kernel $p^D(t, x; s, y)$ of $-(-\Delta)^{\frac{\alpha}{2}}|_D$ which is also the transition density of $X^D$ has the following lower and upper estimates: for every $T > 0$ and $(s, x, y) \in (t, T] \times D \times D$,

$$c_1 \left(1 \wedge \frac{\varrho(x)^{\frac{\alpha}{2}}}{\sqrt{s-t}}\right) \left(1 \wedge \frac{\varrho(y)^{\frac{\alpha}{2}}}{\sqrt{s-t}}\right)$$

$$\leq \quad \frac{p^D(t, x; s, y)}{p(t, x; s, y)}$$

$$\leq \quad c_2 \left(1 \wedge \frac{\varrho(x)^{\frac{\alpha}{2}}}{\sqrt{s-t}}\right) \left(1 \wedge \frac{\varrho(y)^{\frac{\alpha}{2}}}{\sqrt{s-t}}\right),$$

where $\varrho(x)$ denotes the distance between $x$ and $D^c$, the complement of $D$ and $p(t, x; s, y)$ is the transition density defined in example 3. Therefore, under a simple condition on $\alpha$ and $d$, the law $\mu(t, x; s, dy)$ of $X_s^{t,x}$ satisfies ($\mathbf{H}_{4.3}$).

5. For simplicity and ease of notation, we shall take $t = 0$. The Brownian motion killed upon exiting $D$ is defined as

$$W_s^D = \begin{cases} W_s & \text{if} \quad s < \tau_D^W \\ \varkappa & \text{if} \quad s \geq \tau_D^W. \end{cases}$$

Now, we define the subordinate killed Brownian motion, $Y_s^D = W_{A_s}^D$ for all $s \geq 0$, as the process obtained by subordinating $W^D$ via the $\frac{\alpha}{2}$-stable subordinator $A$. That is

$$Y_s^D = \begin{cases} W_{A_s}^D & \text{if} \quad s < \tau_D^W \\ \varkappa & \text{if} \quad s \geq \tau_D^W. \end{cases}$$

Let $r^D(t, x; s, y)$ be the transition density of $Y^D$. It is proved in ([29], Lemma 2.1) that if $D$ is a bounded $\mathcal{C}^{1,1}$ domain in $\mathbb{R}^p$, $p \geq 1$ then for any $T > 0$, there exist two positive constants $C_3$ and $C_4$ such that for any $s \in (t, T]$ and $x, y \in D$.

$$C_3 q^D(s - t, x, y) \leq \frac{r^D(t, x; s, y)}{p(t, x; s, y)} \leq C_4 q^D(s - t, x, y),$$

where

$$q^D(s, x, y) = \left( \frac{\varrho(x)}{(s^{\frac{1}{\alpha}} + |x - y|)} \wedge 1 \right) \left( \frac{\varrho(y)}{(s^{\frac{1}{\alpha}} + |x - y|)} \wedge 1 \right).$$

Therefore, under a condition on $\alpha$ and $p$, the law $\mu(t, x; s, dy)$ of $X_s^{t,x}$ satisfies ($\mathbf{H}_{4.3}$).

6. For $d \geq 2$, we consider the time-inhomogeneous and non-symmetric non-local operators:

$$\mathfrak{L}_t f(x) = \mathfrak{L}_t^a f(x) + b_t \cdot \nabla f(x) + \mathfrak{L}_t^\kappa f(x), \tag{28}$$

where

$$\mathfrak{L}_t^a f(x) = \frac{1}{2} \sum_{i,j=1}^d a_{i,j}(t, x) \frac{\partial^2 f}{\partial x_i x_j}(x), \quad b_t \cdot \nabla f(x) = \sum_{i=1}^d b_i(t, x) \frac{\partial f}{\partial x_i}(x)$$

and

$$\mathfrak{L}_t^\kappa f(x) = \int_{\mathbb{R}^d} \left( f(x + z) - f(x) - \mathbb{1}_{\{|z| \leq 1\}} z \cdot \nabla f(x) \right) \frac{\kappa(t, x, z)}{|z|^{d+\alpha}} dz,$$

where $a(t, x) := (a_{ij}(t, x))_{1 \leq i,j \leq d}$ is a $d \times d$-symmetric matrix-valued measurable function on $\mathbb{R}_+ \times \mathbb{R}^d$, $b(t, x) : \mathbb{R}_+ \times \mathbb{R}^d \longrightarrow \mathbb{R}^d$ and $\kappa(t, x, z) : \mathbb{R}_+ \times \mathbb{R}^d \times \mathbb{R}^d \longrightarrow \mathbb{R}^d$ are measurable functions, and $\alpha \in (0, 2)$.

We denote by $p(t, x; s, y)$ the fundamental solution of the operator $\{\mathfrak{L}_t^a, t \geq 0\}$ and $q(t, x; s, y)$ the fundamental solution of the operator $\{\mathfrak{L}_t, t \geq 0\}$. From (28) $\mathfrak{L}_t$ can be interpreted as a perturbation of $\mathfrak{L}_t^a$ by the operator $b_t \cdot \nabla + \mathfrak{L}_t^\kappa$, so the heat kernels $p$ and $q$ are related by the following Duhamel formula:

$$\begin{aligned} q(t, x; s, y) &= p(t, x; s, y) \\ &+ \int_t^s \int_{\mathbb{R}^d} q(t, x; r, z)(b_r \cdot \nabla + \mathfrak{L}_r^\kappa) p(r, \cdot; s, y)(z) dz dr \\ &= p(t, x; s, y) \\ &+ \int_t^s \int_{\mathbb{R}^d} p(t, x; r, z)(b_r \cdot \nabla + \mathfrak{L}_r^\kappa) q(r, \cdot; s, y)(z) dz dr \end{aligned} \tag{29}$$

for all $0 \leq t < s < \infty$ and $x, y \in \mathbb{R}^d$.

For any $T > 0$ and $\varepsilon \in [0, T)$, we denote

$$\mathbb{D}_\varepsilon^T = \left\{ (t, x; s, y) : s,\ t \geq 0 \text{ and } x,\ y \in \mathbb{R}^d \text{ with } \varepsilon < s - t < T \right\}.$$

It is proved under some mild conditions of the coefficients $a$, $b$, and $\kappa$ (see $(\mathbf{H}^a)$, and $(\mathbf{H}^\kappa)$ in [30], Theorem 1.1) that there exists a unique heat kernel $q(t, x; s, y)$ satisfying (29). Moreover, $q(t, x; s, y)$ is the transition density of the Markov process $X$ associated to the operator $\{\mathfrak{L}_t, t \geq 0\}$. The two-sided estimates below of $q$ were established in ([30], Corollary 1.5): For any $T > 0$, there exist constants $C$, $\lambda \geq 1$ such that on $\mathbb{D}_0^T$:

$$bC^{-1}\left( (s-t)^{-\frac{d}{\alpha}} e^{-\lambda^{-1}\frac{|x-y|^2}{s-t}} + m_\kappa(s-t)\left( (s-t)^{\frac{1}{2}} + |x-y| \right)^{-d-\alpha} \right)$$
$$\leq\ q(t, x; s, y),$$

and

$$q(t, x; s, y)$$
$$\leq\ C\left( (s-t)^{-\frac{d}{\alpha}} e^{-\lambda\frac{|x-y|^2}{s-t}} + \|\kappa\|_\infty (s-t)\left( (s-t)^{\frac{1}{2}} + |x-y| \right)^{-d-\alpha} \right),$$

where $m_\kappa = \inf_{(t,x)} \operatorname{essinf}_{z \in \mathbb{R}^d} \kappa(t, x, z)$.
The law $\mu(t, x; s, \mathrm{d}y)$ of $X_s^{t,x}$ satisfies $(\mathbf{H}_{4.3})$.

## 5. Concluding Remarks

In this paper, we discuss the issues of the global existence of the solutions for a class of multidimensional Markovian backward stochastic differential equations driven by a Poisson random measure and an independent Brownian motion. We first generalized the representation obtained by El Karoui et al. [3] to the jump case which clams that the solution of Markovian BSDEJ with Lipschitz generator can be represented in terms of the Markov process and some deterministic functions. This result, with the help of so-called $L^2$-domination condition, on the law of the underlying Markov process, played a crucial role in proving the main results of this paper. More precisely, we proved that BSDEJ (1) in the case where its generator is continuous with respect to $y$ and $z$ and globally Lipschitz in $k(\cdot)$ has at least a solution. Then, we extended the latter result by allowing the generator to be also continuous in $k(\cdot)$, but only for a particular form of BSDEJ (1). We hope to treat in future research the more general case where the generator of BSDEJ (1) is totally continuous with respect to all its state variables to fill the gaps and solve this open problem.

**Author Contributions:** All authors have contributed equally on the paper. All authors have read and agreed to the published version of the manuscript.

**Funding:** This research was funded by King Saud University grant number RG-1441-339.

**Acknowledgments:** The first and second named authors extend their gratitude to the Deanship of Scientific Research at King Saud University for funding this work through Research Group no (RG-1441-339).

**Conflicts of Interest:** The authors declare no conflict of interest.

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
