# Peer review of "Multidimensional Markovian BSDEs with Jumps and Continuous Generators"

_axioms, doi:10.3390/axioms12010026_

Round 1

Reviewer 1 Report

see attachment. 

Author Response

kindly see the attached file 

Reviewer 2 Report

The paper must contain an applied example that demonstrates the importance of this paper, as well as there must be numerical examples that demonstrate the behavior and stability of the proposed system

Author Response

Kindly see the attached file

Reviewer 3 Report

This study analyses a multidimensional Markovian backward stochastic differential equation driven by a Poisson random measure and independent Brownian motion (BSDEJ for short). As a first result, the authors prove, under Lipschitz condition, that the BSDEJ’s adapted solution can be represented in terms of a given Markov process and some deterministic functions.

The subject of the manuscript is interesting. It contains new results and can be considered for publication after minor revision considering the following points: 

The abstract should be more prominent in which authors should include important findings. 

Please add the conclusion section.

This paper should be edited grammatically.

Author Response

Kindly see the attached file

Reviewer 4 Report

If the goal of the paper is interesting, there are too much troubles in the paper. I think that the framework has to be clarified (and simplified) and the mathematical issues have to be corrected. Details can be found in the attached file. 

Author Response

Kindly see the attached file

Round 2

Reviewer 4 Report

See attached file

Author Response

Thank you very much for your interesting remark. 

Kindly find a response in the attached file. 
